# *Cis*-Acting Factors Causing Secondary Epimutations: Impact on the Risk for Cancer and Other Diseases

**DOI:** 10.3390/cancers13194807

**Published:** 2021-09-26

**Authors:** Miguel Ruiz de la Cruz, Aldo Hugo de la Cruz Montoya, Ernesto Arturo Rojas Jiménez, Héctor Martínez Gregorio, Clara Estela Díaz Velásquez, Jimena Paredes de la Vega, Fidel de la Cruz Hernández-Hernández, Felipe Vaca Paniagua

**Affiliations:** 1Laboratorio Nacional en Salud, Diagnóstico Molecular y Efecto Ambiental en Enfermedades Crónico-Degenerativas, Facultad de Estudios Superiores Iztacala, Tlalnepantla 54090, Mexico; miguel.ruiz.cruz@cinvestav.mx (M.R.d.l.C.); erarroji@gmail.com (E.A.R.J.); mag_hector@hotmail.com (H.M.G.); cdiazvelasquez@aol.com (C.E.D.V.); jim.paredes@hotmail.com (J.P.d.l.V.); 2Avenida Instituto Politécnico Nacional # 2508, Colonia San Pedro Zacatenco, Delegación Gustavo A. Madero, C.P. Departamento de Infectómica y Patogénesis Molecular, Centro de Investigación y de Estudios Avanzados del Instituto Politécnico Nacional (CINVESTAV-IPN), Mexico City 07360, Mexico; cruzcruz@cinveztav.mx; 3Unidad de Biomedicina, Facultad de Estudios Superiores Iztacala, UNAM, Tlalnepantla 54090, Mexico; audelacm@gmail.com; 4Subdirección de Investigación Básica, Instituto Nacional de Cancerología, Ciudad de México 14080, Mexico

**Keywords:** epigenetic, epimutations, promoter hypermethylation, secondary epimutations, *cis*-acting factors

## Abstract

**Simple Summary:**

Epigenetic mechanisms contribute to the regulation of gene expression. However, when they fail, they result in diseases such as cancer. Among these effects is aberrant DNA methylation caused by inherited mutations in *cis* of the affected gene, referred to as constitutional secondary epimutations. Little is known about this phenomenon, in which hypermethylation promotes transcriptional silencing of tumor suppressor genes in patients with inherited cancers that do not have pathogenic variants in the coding region of cancer susceptibility genes. Here we discuss these hereditary alterations and their effect during the early stages of tumorigenesis, as well as their contribution to disease historically and from a molecular perspective.

**Abstract:**

Epigenetics affects gene expression and contributes to disease development by alterations known as epimutations. Hypermethylation that results in transcriptional silencing of tumor suppressor genes has been described in patients with hereditary cancers and without pathogenic variants in the coding region of cancer susceptibility genes. Although somatic promoter hypermethylation of these genes can occur in later stages of the carcinogenic process, constitutional methylation can be a crucial event during the first steps of tumorigenesis, accelerating tumor development. Primary epimutations originate independently of changes in the DNA sequence, while secondary epimutations are a consequence of a mutation in a *cis* or *trans*-acting factor. Secondary epimutations have a genetic basis in *cis* of the promoter regions of genes involved in familial cancers. This highlights epimutations as a novel carcinogenic mechanism whose contribution to human diseases is underestimated by the scarcity of the variants described. In this review, we provide an overview of secondary epimutations and present evidence of their impact on cancer. We propose the necessity for genetic screening of loci associated with secondary epimutations in familial cancer as part of prevention programs to improve molecular diagnosis, secondary prevention, and reduce the mortality of these diseases.

## 1. Introduction

Cancer is caused by genetic, metabolic, inflammatory, and epigenetic factors [1]. The abnormal proliferation, the major characteristic of cancer, begins with the progressive accumulation of multiple mutations that provide evolutionary adaptations to the tumor cells. Although epigenetic mechanisms were initially overlooked, it is now known that they contribute importantly to cancer development by their capacity to alter the gene expression independently of mutations [2] (Figure 1). These epigenetic mechanisms can be classified into three main molecular groups: non-coding RNAs (ncRNAs), histone post-translational modifications, and DNA/RNA methylation [3,4].

Deregulation of ncRNAs is one of the main regulatory epigenetic mechanisms altered during carcinogenesis and cancer progression [4]. The three most studied classes of ncRNAs are miRNAs, lncRNAs, and circRNAs, which are involved in complex interactions during cancer initiation and promotion [5]. Potent tumorigenic effects of miRNAs have been well described in different malignancies and polymorphisms in these genes may cause cancer susceptibility [6,7]. LncRNAs have been shown to aberrantly promote metastatic pathways and specific mutations in these genes have a key cancer role [8]. In addition, circRNAs have oncogenic activity in organ-specific cancers, neurodegenerative pathogenic conditions, diabetes, and cardio-vascular disease [9]. For a more detailed discussion on the oncogenic role of ncRNAs we refer the reader to the comprehensive reviews published elsewhere [10,11,12].

Histone post-translational modifications alter chromatin architecture and their spatial distribution. The mechanisms that cause these changes operate at the level of both nuclear topology and histone modifications driven by the activity of the Polycomb (PcG) and Trithorax (TrxG) chromatin-modifying complexes [3,13]. Nuclear topology importantly influences the mutational composition in cancer cells, as evidenced by mutations acquired by smoking and UV light enriched at the nuclear periphery [14,15]. PcG and TrxG complexes regulate modifications on histone amino tails through repression signals (polycomb) and gene activation marks (trithorax), changes that collectively have an important role in cellular processes such as apoptosis and senescence [16,17]. Notably, there are more than 60 different histone chemical marks, including methylation, acetylation, phosphorylation, citrullination, sumoylation, adenosine diphosphate ribosylation, deamination, and crotonylation, which regulate the chromatin structure and locus accessibility, improve gene expression in euchromatin regions or mediate heterochromatin gene repression [18].

DNA methylation is the only epigenetic mechanism that directly affects the chemical structure of DNA regulating gene expression through stable silencing. Technical molecular advances for the detection of DNA methylation have given access to a wide range of studies in the fields of embryology, organism development, and disease, by global 5meC quantification [19], locus-specific DNA methylation approaches (sodium bisulfite modification strategies [20,21,22] and methylation-sensitive restriction enzyme sequencing [23,24]), and genome-wide DNA methylation [25,26,27], providing a detailed profile on genome-wide DNA methylation. No technique excels in all aspects. The number and characteristics of the sample as well as the desired precision, coverage, and resolution, define the choice of the appropriate technique. Initial methylation analyzes used qualitative approaches such as MSP, MeDIP-PCR, and microarrays, and focused only on the presence or absence of methylation in a specific locus or CpG islands. Over time, novel methylation techniques such as pyrosequencing and next generation sequencing have been coupled with an initial bisulfite conversion treatment. This has allowed to analyze, at a whole genome level, a wide range of DNA methylation patterns and provides quantitative values on an absolute scale of β distribution from 0 to 1, or 0 to 100%. For more details on methods for DNA methylation analysis, the reader is referred to the review of Laird (2010) [28]. These technological advances in DNA methylation analysis contributed to making this mark the most studied epigenetic modification in humans.

DNA methylation is associated with repressive histone modifications and the interaction between these epigenetic variations is crucial to regulate the function of the genome by changing the architecture of chromatin [29]. In this regard, methylation involves a post-replicative chemical modification through the transfer of a methyl group, mainly in CpG dinucleotides in mammals, which comprise approximately 3–6% of the total cytosines of human DNA [30], although methylation in N6-adenosine has also been described [31]. The enzymes that catalyze methylation in CpG dinucleotides are DNA methyltransferases (DNMTs). Methylation is stable through cell division, can be added de novo, and is also reversible [30,32,33]. There are three active DNMTs in mammals responsible for methylation, of which DNMT1 participates in the maintenance of the methylation patterns during cell division. Conversely, DNMT3A and DNMT3B carry out the de novo methylation in response to molecular stimuli. An additional participant in the DNA methylation process is DNMT3L, a regulatory protein that does not have methyltransferase activity, but stabilizes the DNMT3A/B-DNA complex to increase de novo methylation [34,35]. DNA methylation plays an extremely important role throughout the body, as demonstrated by its involvement in crucial biological processes including regulation of gene expression [36], early development [37], protection against intragenomic parasites [38], genomic imprinting [39], and the inactivation of chromosome X [40]. Consequently, alterations in this regulatory mechanism lead to an aberrant increase (hypermethylation) or decrease (hypomethylation) in basal levels that have been associated with diseases such as cardiovascular problems [41,42], mental disorders [43,44], and different types of cancer [45,46,47]. 

When these mechanisms cause aberrant disruptions in DNA activity, they are called epimutations [48]. Epimutations are epigenetic alterations that result in transcriptional repression or activation of a gene that is normally active or repressed, respectively [49,50]. They occur when a paternal or maternal allele has >50% aberrant methylated or demethylated CpG sites, showing not successful methylation imprinting [51]. Epimutations are grouped into two types [52]: (i) primary epimutations, caused by adaptive aberrant molecular mechanisms that do not involve the alteration of the DNA sequence, and (ii) secondary epimutations, acquired as a consequence of DNA mutations in *cis* or *trans*-acting factors [48].

In this article, we review the relationship between epimutations that alter gene expression and disease. We especially focus on secondary constitutional epimutations caused as a result of inherited DNA *cis*-alterations and describe their experimental evidence over time.

## 2. Origin and Consequences of Epimutations

Mutational events that contribute to carcinogenesis may occur during gametogenesis and embryonic development (Figure 2). Somatic mutations are alterations in the DNA sequence in any cell of the body that take place after fertilization. Hence, they are not passed onto the offspring, but tissues derived from the mutated cell are affected (Figure 2A) [53]. On the other hand, germline mutations are DNA changes present in gametes that can be inherited to the offspring affecting all cells throughout the body and half of the offspring’s gametes (Figure 2B) [54].

On a different level, epimutations are epigenetic alterations that can occur at specific stages in normal cells, for example, when a tumor suppressor gene is methylated during the early stages of embryonic development (Figure 2C). Recently, it has been reported that epimutations could act as initial events of tumorigenesis and together with somatic and germline mutations lead to malignant transformation and progression [55]. 

Epimutations studies are based on genomic imprinting, a methylation state that establishes parent-specific allele expression. A pathological phenotype could be expressed if imprinting is deficient even in one allele. The genomic-specific phenotypes of imprinting are regulated by imprinting centers (ICs). These ICs are short sequences in *cis* of an imprinted gene that regulate parent-specific gene expression bidirectionally over long distances in which only one allele is methylated and transcribed, allowing for imprinting regulation and allele-specific expression. Therefore, genetic alterations in the sequences of ICs can lead to constitutional epimutations [52]. Constitutional epimutations are monoallelic and can be found in tissues from the three germ layers: endoderm, ectoderm, and mesoderm [56]. 

During early human development, the epigenetic modifications throughout the genome occurs dynamically. At this stage, the reprogramming of methylation has functions in the imprinting, the control of gene expression, and the establishment of the cell lineage [57]. Several studies have confirmed that during embryonic development the epigenetic marks of the parents (F_0_) are erased through an active demethylation mechanism [58,59]. Later, during fertilization, the differentially methylated germ regions (gDMR) are established by de novo methylation, including ICRs. During this stage, constitutional epimutations are more likely to be generated. In the zygote, there is a selective demethylation phenomenon of active demethylation of the paternal genome followed by passive demethylation of the maternal genome within the preimplantation embryo [60]. It has been proposed that this selective demethylation results in the specificity of imprinted genes. However, when it fails, it could result in the impairment of methylation in the imprinted loci, generating epimutations throughout the soma. Therefore, the partial “erasure” of a constitutional epimutation in a proportion of cells in the pre-implanted embryo could result in somatic mosaicism of the epimutation in the adult [61,62]. In addition, a de novo methylation was generated in the inner cell mass (ICM) of the blastocyst and has an important role in early lineage establishment. During this stage of the early development, constitutional epimutations could be generated [58]. 

In the 1990s, epimutations in tumor suppressor genes such as *RB* [63], *VHL* [64], *MLH1* [65], and *BRCA1* [66] were identified in sporadic cancers. However, over time, the research about this phenomenon in hereditary cancers has been strengthened. We discuss the role of epimutations in inherited cancers in the following section.

## 3. Aberrant Methylation as the Initial Step of Carcinogenesis in Hereditary Cancers

Epimutations are one of the early molecular alterations in tumorigenesis and may be a direct cause of carcinogenesis [67]. The causality in cancer is associated with common and uncommon genetic variants [58]. However, these variants only explain a portion of the individuals with the phenotype. On the other hand, epigenetics has been established as an alternative mechanism to explain causality in cancer [68].

In 1971, Knudson postulated the “two-hit hypothesis” to explain how two mutations or “hits” were required in both *RB1* alleles to develop retinoblastoma [69]. Nowadays, it is used to explain the pathogenesis of tumors in autosomal dominant cancer syndromes [70]. Knudson’s hypothesis requires that both alleles of a tumor-suppressing gene must be impaired to cause cancer. The first hit frequently comes from germline mutations, the second hit may result from somatic mutations such as SNV or aberrant methylation of the second allele [71] (Figure 3). Interestingly, constitutional epimutations can either act as the first hit in concordance with the Knudson’s model to develop cancer [72,73], or as the second hit in some tumors associated with familial cancer syndromes caused by heterozygous germline mutations [74]; contributing in this way to the carcinogenic process. Accordingly, germline epimutations that initiate during embryonic development could contribute to the “first hit” in tumorigenesis in cancer susceptibility genes. Some alterations favor transcriptional silencing and act as the first hit. Some examples of these mechanisms include the direct impairment of DNA binding of transcription factors (TFs) to the promoter region [75], the backward movement of methyl-CpG (MBD) binding domain proteins in methylated DNA, which compete directly with TFs through their greater affinity for the 5-mC site in the DNA sequence [76], and the synergistic binding effect of transcriptional repression factors (TFRs), which bind with affinity to DNA when an SNV is present. This phenomenon will be addressed in the Section 5 [77].

One of the first pieces of evidence that associated the two-hit hypothesis and epimutations was a study conducted on the *MLH1* gene. A seminal study by Gazzoli et al. described an epimutation that caused *MLH1* promoter hypermethylation. Interestingly, this epimutation was detected in one allele (from peripheral blood DNA) in a patient with colorectal cancer, and the second unmethylated allele was inactivated due to loss of heterozygosity in the tumor tissue. This biallelic inactivation resulted in the complete loss of MLH1 expression in the tumor, confirmed by immunohistochemistry. These findings suggested a new mechanism of germline inactivation in a cancer susceptibility gene [78]. A second work confirmed this aberrant mechanism in *MLH1* in two individuals who did not have a germline mutation in DNA repair genes but presented clinical criteria for hereditary colorectal cancer; concluding that the germline epimutation in *MLH1* leads to somatic mosaicism consistent with epigenetic states that resemble those of polygenic or complex traits [79].

## 4. Primary Epimutations and Environmental Factors

The epigenetic architecture of cells and tissues is actively shaped by their environment and molecular context. The molecular factors that influence the epigenetic landscape of cells are stress, proliferative and cell cycle regulatory signals, specific extracellular molecules, nutrition, among others [80].

Primary or “true” epimutations are aberrant adaptive changes in response to molecular stimuli that occur without any change either in the DNA sequence or in ICs, but as a result of genetic, environmental, and stochastic factors that lead to transcriptional repression (Figure 4(AI)) [52]. One of the best-characterized examples of a primary epimutation occurs in patients with Wilms tumors, where the maternal allele of the imprinting control regions (ICR) upstream of the *H19* gene is aberrantly methylated, leading to loss of the imprinting (LOI) [81].

The environmental factors that have been linked to the modification of the methylation patterns include exposure to chemical agents such as heavy metals [82], aerosols in the air [83], cigarette smoke [84], endocrine-disrupting substances, or non-genotoxic carcinogens [85]. All these factors can generate changes in gene expression, activation of signaling pathways, and alterations in the epigenome and transcriptome [86].

The nutritional status is an environmental factor that is associated with primary epimutations that affect the methylation pattern [71], as well as the risk of cancer [87,88]. Dietary components rich in folates, choline, and vitamin B12 are known to influence the methylation status by increasing the levels of SAM (S-adenosyl-methionine), exhibiting a direct link between diet and the epigenetic state [89,90]. SAM serves as an enzymatic substrate in which DNA methyltransferases obtain the methyl groups to bind to DNA [91]. The impact of dietary folate on changes in DNA methylation has been observed in breast, prostate, stomach, colon, and thyroid cancers [92], as well as in animal models [93]. Thus, when evaluating the effects of the absence of folates in a rat model supplemented with vitamin B12 deficiency, DNA hypomethylation was found in colon tissue, which indicates that the restricted diet in the absence of B12 greatly reduces DNA methylation [94]. 

Also, primary epimutations have been linked to aging and may contribute to the late onset of cancer [95]. For example, during the aging process, DNA hypermethylation is facilitated competitively due to the destabilization of the repressive polycomb complex [96].

## 5. Secondary Epimutations

Secondary epimutations are driven by *cis* or *trans*-acting genetic alterations (Figure 4(AII)) [52]. Secondary epimutations that operate through the action of factors in *trans* include DNA mutations in genes that encode enzymes involved in DNA methylation, histone acetylation or deacetylation, histone methylation, chromatin remodeling, and epigenetic mark recognition [97]. However, secondary epimutations of *cis* action include deletions, insertions, substitutions, and changes in the length of tandem repeated sequences known as copy number variations (CNV) [98], all capable of affecting ICs [49]. Notably, these alterations can experience the following types of inheritance patterns: (i) autosomal dominant transmission of epimutations caused by genetic *cis*-acting alterations [55]; (ii) “Null” inheritance due to sporadic de novo epimutations with concomitant germ-cell erasure of epimutations and re-establishment of the somatic aberrant state in the next generation [99,100] and (iii) non-Mendelian transmission or incomplete penetrance [99]. This translates into a complex and heterogeneous landscape in the inheritance mechanisms of epimutations, even when the same gene is affected (Figure 4B). These findings were discovered on the *MLH1* gene and probably additional loci may show these inheritance patterns. 

In addition to secondary epimutations caused by *cis*-acting factors, secondary epimutations caused by mutations in *trans*-acting factors include alterations in the coding region of DNA methyltransferases, methyl-CpG binding proteins, histone-modifying enzymes, and the chromatin remodeling complex, among others (Figure 4(AII)). These protein complexes are essential for epigenetic maintenance and its deregulation generates a critical panorama for diseases and a potential early death of the patient [52]. One example is the case of de novo DNA methyltransferase (DNMT3A) mutations, which cause the autosomal recessive ICF syndrome (immunodeficiency, centromere instability, and facial anomalies). Patients with this syndrome present microsatellite instability associated with severe DNA hypomethylation, resulting in death at a young age from recurrent serious infections. Other diseases caused by secondary epimutations are mental retardation syndromes with X-linked–α-thalassemia (ATRX) and Rett syndrome (RTT). The former is a disorder characterized by mental retardation, facial dysmorphia, genital abnormalities, alpha thalassemia, and it is caused by a reduced expression of the α-globin genes that encode a member of the SWI/SNF complexes, which plays an important role in the system of chromatin remodeling and DNA repair [101]. The RTT is a rare genetic neurological disorder that affects brain development and causes severe mental and physical disability caused by mutations in the *MECP2* gene, which encodes the methyl-CpG2 binding protein. MECP2 recruits the corepressor complex SIN3A containing histone deacetylase that favors the repressive chromatin state, leading to transcriptional inactivation [52,102].

## 6. *Cis*-Acting Factors Causing Secondary Epimutations: Historical Evidence

Constitutional secondary epimutations with *cis* effect have been described in several loci, but particularly in tumor suppressor genes including *FMR1, SNURF/SNRPN, HBA2, H19, LIT1, MSH2, DAPK1, MLH1, CDH1, MGMT, MMACHC,* and *BRCA1.* During the 1990s, the first reports were focused on promoter *cis*-acting epimutations. CGG repeat expansions of more than 200 trinucleotide repeats, located at the 5’UTR end of fragile X mental retardation gene 1 (*FMR1*) causes a neurodevelopmental disorder called the fragile X syndrome (FXS) (Figure 5A) [103,104]. To better understand the role of these repeated CGG expansions, subsequent studies evaluated cytosine methylation patterns within a 220 bp region on the CpG island of the human *FMR1* gene in peripheral blood, finding hypermethylation in this region [104]. These initial studies demonstrated that *FMR1* is also methylated and inactivated after the expansion of CGG repeats in fragile X syndrome. Methylation variability was most pronounced in a consensus binding sequence for the transcription factor α-PAL, a sequence that may play a role in regulating *FMR1* expression. This evidence suggests that the maintenance of cytosine methylation is a dynamic process and could generate a fragile X-male methylation mosaicism [105].

As mentioned before, epimutations are involved in genomic imprinting. Several genetic alterations have been found in ICs such as microdeletions that result in secondary epimutations in both Prader-Willi (PWS) and Beckwith-Wiedemann syndromes (BWS) [67]. One of the genes of importance for PWS is *SNURF* (*SNRPN* upstream reading frame protein), which is found within a critical region of chromosome 15. The transcripts of this gene are initiated in an IC and have paternal imprinting [106,107]. These transcripts can be bicistronic and encode SNRPN (small nuclear ribonucleoprotein polypeptide N) from a downstream open reading frame. These regions were initially mapped at 100 Kb (including exon 1) [108], detecting its differential methylation at the CpG island of *SNRPN* [109,110,111,112] and intron 7 [109,113,114,115]. However, it has been reported that deletions in the paternal allele in the PWS lead to an epigenetic state that resembles the maternal imprinting [116,117]. PWS clinical characteristics include muscular hypotonia, hypogonadism, obesity, short stature, and mild to moderate mental retardation. One study of 51 patients with PWS showed that seven patients (14%) presented a paternally inherited deletion in the IC region affecting 4.3 Kb around exon 1 of *SNURF-SNRPN* [116,117]. This alteration influenced the maintenance of paternal imprinting during early embryogenesis, eventually leading to PWS (Figure 5B). These microdeletions have been named Prader-Willy Syndrome Smallest Region of deletion Overlap (PWS-SRO) for their association with the PW syndrome.

One interesting study of a patient with a family history of α-thalassemia showed no mutations in the exonic sequence of the *HBA2* gene. However, the promoter of one allele was found to be hypermethylated and transcriptionally silenced in peripheral blood lymphocytes, with the absence of methylation in sperm. This constitutional secondary epimutation was caused by an 18 kb deletion downstream of *HBA2*, which eliminates some genes including *LUC7L*; that is antisense transcribed in the opposite direction to the *HBA2* gene [118,119]. This deletion eliminates three exons of the *LUC7L* gene, including the stop codon. Its antisense transcription extends across the CpG island of the *HBA2* promoter, causing its hypermethylation and the complete allelic silencing of the *HBA2* gene (Figure 5C) [120].

**Figure 5 cancers-13-04807-f005:**
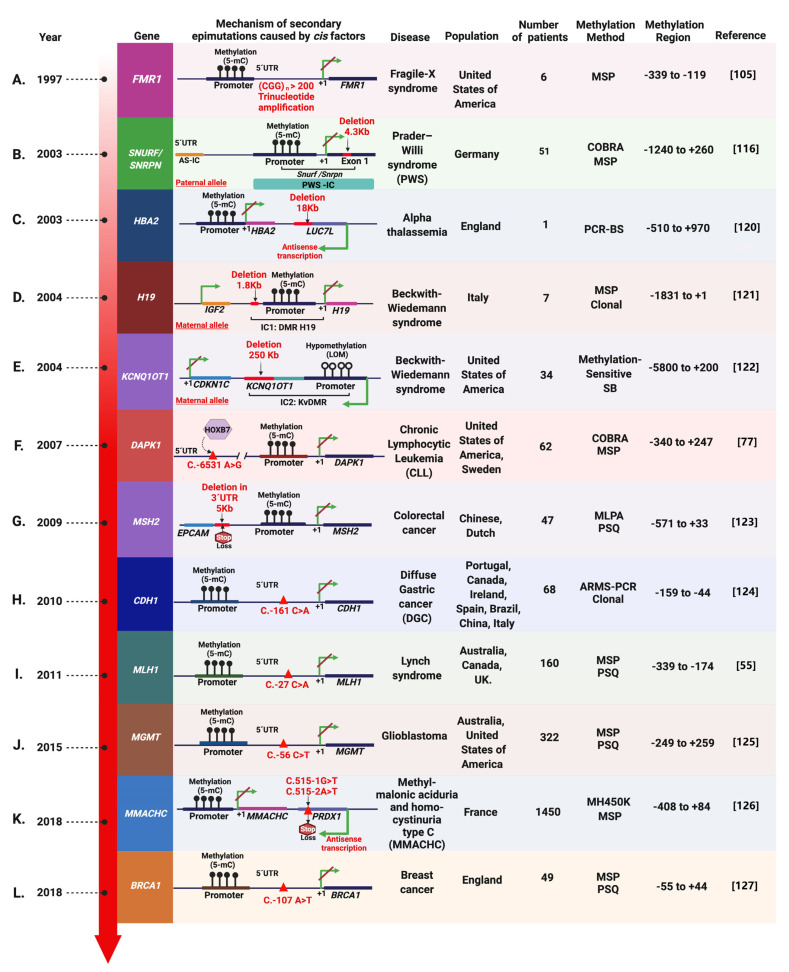
Chronological reports of secondary epimutations causing promoter methylation in cancer and other diseases [55,77,105,116,120,121,122,123,124,125,126,127]. Red color indicates the *cis*-acting factor that results in secondary epimutation. The details of each disease and their pathogenic secondary epimutations are described in the text. Green arrows indicate the site +1 (ATG) of the gene and red diagonals indicate transcriptional repression of the gene. The methylation region position is described relative to the transcription start site. Methods: MSP, Methylation-Specific-PCR; PSQ, Pyrosequencing. COBRA, Combined Bisulfite Restriction Analysis; PCR-BS, PCR-Bisulfite sequencing; MLPA, Multiplex Ligation-Dependent Probe Amplification; ARMS-PCR, amplification refractory mutation system-based PCR.

Another disease associated with aberrant genomic imprinting is the Beckwith-Wiedemann syndrome (BWS); a congenital overgrowth disorder, whose most frequent clinical characteristics are microglossia (97%), abdominal wall defect (80%), birth weight, or postnatal growth above 90th percentile (88%), ear folds/pits (76%), nephromegaly (59%) and hypoglycemia (63%) [128,129,130]. Furthermore, 5–10% of patients with BWS have been reported at high risk for developing malignant tumors, including Wilms tumor, hepatoblastomas, rhabdomyosarcomas, and adrenocortical carcinomas [131,132]. Alterations in the 11p15.5 locus cause BWS [128]. This genomic region has two imprinted domains: IC1 that regulates the expression of insulin-like growth factor 2 (*IGF2*) and *H19*; and IC2 controls cyclin-dependent kinase inhibitor 1C and *KCNQ1* with antisense transcript 1 (*CDKN1C/KCNQ1OT1*). Epimutations of IC1 with hypermethylation of the maternal allele in the *H19* DMR (differentially methylated region of the *H19* gene) account for 5% of BWS cases (Figure 5D). This epimutation results in the biallelic expression of *IGF2* and silencing of *H19* expression [128]. These findings were strongly complemented by the investigations of Sparago et al., identifying two families that showed a microdeletion in IC1: *IGF2/H19.* Transmission of the deletions in the maternal allele resulted in hypermethylation of IC1: *H19*, biallelic expression of *IGF2*, silencing of *H19,* and BWS [121,133].

Secondary epimutations in the IC2 region with loss of methylation (LOM) in KvDMR1 (differentially methylated region of the *KCNQ1OT1* gene) represent less than 2% of BWS cases (Figure 5E). This epimutation in the maternal allele results in the biallelic expression of *KCNQ1OT1* (also known as *LIT1*) and the silencing of *CDKN1C* (also known as *P57^Kip2^)* in both alleles [128,134,135,136]. Subsequently, it was demonstrated by Niemitz et al., that aberrant maternal expression of *LIT1* is conditioned by a microdeletion in IC2: KvDMR1 spanning approximately 250 kb and including almost all of the *LIT1* coding region [122].

The presence of secondary epimutations in Chronic Lymphocytic Leukemia (CLL) has been widely documented. CLL is characterized by uncontrolled proliferation of lymphocytes (lymphocytosis) whose complications may lie in splenomegaly, anemia, and hypogammaglobulinemia, leading to recurrent infections. Between 5–10% of CLL cases are familial [137] and the silencing of death-associated protein kinase 1 (*DAPK1*) has been strongly associated with an epimutation mechanism. *DAPK1* was identified as a familial tumor suppressor gene and it has shown aberrant promoter hypermethylation in CLL (Figure 5F). This secondary epimutation was associated with an SNV (c.-6531A>G) approximately 65 Kb upstream of the *DAPK1* gene and has a dominant inheritance pattern [62]. Additionally, functional assays revealed a decrease of transcriptional activity directly associated with a higher binding affinity of the homeobox transcriptional repressor (*HOXB7*) in the DAPK1 promoter in the pathogenic allele [77].

Evidence linking constitutional epimutations to cancer predisposition has been well documented in some conditions like Lynch syndrome [81,138]. This syndrome has a dominant inheritance pattern and is characterized by an elevated risk to develop colorectal, endometrial, and other types of cancer [139]. It is caused by germline mutations in DNA mismatch repair (MMR) genes, including *MLH1*, *MSH2, or MSH6* causing microsatellite instability (MSI) [140] in 2–3% of all colorectal cancers [141,142]. The majority of cases with high MSI have somatic hypermethylation of the *MLH1* promoter [143]. MSI in patients with Lynch syndrome or spontaneous cancer with *MLH1* promoter hypermethylation indicates that epimutations as well as somatic mutations affecting *MLH1* are early onset in the carcinogenesis of colorectal cancer. Patients with clinical suspicion of Lynch syndrome, but without a germline mutation in an MMR gene or a familiar history of cancer, typically show aberrant methylation of the *MLH1* promoter in normal and tumor tissues, suggesting that it is a causative agent for cancer development in these people [72]. Dominant transmission of an *MLH1* epimutation with somatic mosaicism and transcriptional repression linked to a particular genetic haplotype was reported in a family affected by colorectal cancer [55,144,145]. Epimutation was erased in sperm, but was reestablished in somatic cells of the next generation, consistent with an inherited genetic effect causing the secondary hypermethylation phenotype. The affected haplotype consisted of a single nucleotide substitution, c.-27C>A, located near the transcription start site (Figure 5I). Importantly, functional assays to test this variant showed a significantly reduced transcriptional activity and is likely to be the cause of this secondary epimutation [55,146]. 

Furthermore, one of the most important studies over time is one of the constitutional epimutations in the *MSH2* gene and their role in colorectal cancer development. It was first described in 2006 by Chan et al. [147], where a three-generation Chinese family with hereditary nonpolyposis colorectal cancer (HNPCC) exhibited germline allele-specific hypermethylation in the *MSH2* promoter region without having mutations in DNA mismatch repair genes (MMR); this was key to explain the etiology of Lynch syndrome in the family. Three family members who were carriers of germline epimutations developed early-onset endometrial or colorectal cancer, all with evidence of MSI and allelic loss of *MSH2* [147,148]. In the following three years, Ligtenberg et al. showed that this secondary epimutation of the *MSH2* promoter region was a consequence of a 5 Kb deletion at the 3’ end of a neighboring *EPCAM* gene upstream of *MSH2*, located at 17 Kb (Figure 5G). This *EPCAM* deletion includes the polyadenylation signal that results in a continuous transcription, leading to an *EPCAM-MSH2* fusion transcript and transcriptional repression of *MSH2* [123,149]. Gastric cancer (GC) studies have shown great advances in the field of epimutations. The hereditary familial form of GC is known as diffuse hereditary gastric cancer (HDGC), has an autosomal dominant pattern, and is responsible for 1% of all types of GC. Despite its low frequency, HDGC is a public health problem due to its severity and late diagnosis [150,151]. In a study of proband patients with HDGC disease with the absence of germline mutations, hypermethylation of the *CDH1* gene promoter (Cadherin-1) in peripheral blood DNA was found in a single patient. Aberrant methylation was conditioned by a specific allele with rs16260 (C.-161C>A) SNP (Figure 5H). 

However, it was not possible to identify the epimutation in any other member of the family besides the proband because they had died due to the disease, which made it impossible to clarify the pattern of the inheritable epimutation of gastric cancer [124]. 

The gene *MGMT*, which codes for the DNA repair enzyme O^6^-methylguanine-DNA methyltransferase (MGMT) has also been studied at the epigenetic level. The MGMT enzyme eliminates the transitional mutation of O^6^ methylguanine and participates in therapy resistance to alkylating agents [152]. The activity of *MGMT* is regulated by its promoter and its hypermethylation leads to gene silencing in cancer [153,154]. Epimutations in *MGMT* have been linked to various types of cancer, such as colorectal cancer, lung cancer, and brain gliomas [125,153,154,155,156,157]. An epimutation in the promoter region of the *MGMT* gene is one of the most important prognostic factors for patients with glioblastoma and it can predict the response to treatment with alkylating agents such as temozolomide [155]. However, it has been determined that the SNP rs16906252 (C.-56C>T) found in one of the parental alleles, is a determinant key in the acquisition of the *MGMT* epimutation in glioblastoma (Figure 5J). Additionally, it is known that temozolomide-treated patients with the rs16906252 T genotype have better survival probabilities, regardless of tumor methylation status [125,158].

Hereditary secondary epimutations in non-oncologic syndromes have also been reported. That is the case of the Methyl-Malonic Aciduria and Homocystinuria type cblC (*MMACHC*) gene, which was recently studied by Guéant et al., in 2018. This research group identified a cause of the cblC class (cobalamin, cblC), an autosomal recessive disease characterized by inborn errors in vitamin B12 metabolism, named “epi-cblC”. The cblC disorder presents both neurological and systemic metabolic abnormalities [159]. Patients with epi-cblC are heterozygotes for a genetic mutation and have a secondary epimutation at the *MMACHC* locus, which is flanked by the *CCDC163P* and *PRDX1* genes and oriented in the opposite direction. Secondary epimutation was triggered by mutations in the neighboring antisense gene *PRDX1* that produces an aberrant antisense transcript leading to loss of the stop codon (Figure 5K). Interestingly, this epimutation was found in three generations with *PRDX1* mutations in *cis* (c.515-1G>T, c.515-2A>T) that strengthen the antisense transcription of MMACHC and possibly triggering the activation methylation mark H3K36me3. Furthermore, this work demonstrated that the silencing of *PRDX1* transcription leads to partial epiallele hypomethylation and restoration of the expression of *MMACHC* [126]. 

Secondary epimutations have also been evaluated in breast and ovarian cancer, where people who carry germline *BRCA1* and *BRCA2* mutations are at high risk of developing this syndrome. In sporadic breast and ovarian tumors, epimutations of the *BRCA1* promoter are well-known players of somatic carcinogenesis [160,161]. However, recently, Evans et al. (2018) identified a dominant inherited 5’UTR variant associated with epigenetic silencing of *BRCA1* due to promoter hypermethylation in two families affected by breast and ovarian cancer. The clinical condition of the two families was consistent with hereditary breast and/or ovarian cancer (HBOC) but they lacked pathogenic *BRCA1* or *BRCA2* variants. The secondary epimutation was *BRCA1* promoter hypermethylation of 10 CpG sites in these families. Hypermethylation of the *BRCA1* promoter was detected in 2 of 49 independent families in the United Kingdom. The proband women were affected by high-grade breast cancer, and the hypermethylation of the *BRCA1* promoter was detected in blood, oral mucosa, and hair follicle tissues. Moreover, RNA sequencing revealed the allelic loss of *BRCA1* expression in both families and its cosegregation with the heterozygous variant c.-107A>T at the 5’UTR end of the *BRCA1* gene (Figure 5L) [127]. Laner et al. conducted a study to detect the presence of this same variant using a competitive allele-specific PCR assay (KASP) or direct Sanger sequencing in 3297 German patients with HBOC criteria without pathogenic germline variants. In their results, they did not detect any individual carrying the variant. Therefore, they concluded that in *BRCA1*, the variant c.-107A>T is not common in the population of southeast Germany [162].

Research efforts are also currently directed at the improvement of epigenetic anomalies based on population studies, called “epigenetic epidemiology”. These studies are progressing in great strides by including next-generation technologies and covering a large number of patients to identify these “epivariants” or “epitypes” that condition the phenotype of the disease. It is the case of the robust study by Garg et al., 2020, on the methylation profiles of 23,116 individuals using the Illumina 450k matrix. They identified more than 4000 unique autosomal patterned epivariants potentially leading to promoter epimutations in more than 300 human disease-related tumor suppressor genes. This work suggests that epivariants may underlie a fraction of human disease that would be missed with purely genetic sequence-based approaches. This study provides a broad overview of rare epigenetic changes in the human genome, giving insight into the underlying origins and consequences of epivariations, potentially altering the methylation pattern related to human diseases [163]. In this light, it is important to distinguish the constitutional secondary epimutations that cause diseases from the common epigenetic variants, usually influenced by ethnicity and caused by non-pathogenic natural adaptive mechanisms in humans [164].

## 7. Conclusions

The pathogenic role of constitutional inherited secondary epimutations has substantial evidence. The *c**is* pathogenic effect on epimutations is wide and affects imprinted genes, cancer-related genes, and other hereditary syndromes. However, as it is a rare mechanism, the pathogenic allele frequencies of these alterations among patients and the general population are still unknown and underestimated due to the absence of epigenetic monitoring of patients. Therefore, there is still a long way to go for epimutations to be included in diagnostic procedures and incorporated into the molecular detection strategies of hereditary cancers. Even in the case of the *MLH1* and *MSH2* genes where secondary epimutations have an established pathogenic role in disease, epigenetic screening has not yet been implemented as a routine test.

Therefore, it is key to thoroughly describe, determine the population frequencies, and understand the inheritance mechanisms of secondary epimutations, an effort that will undoubtedly contribute to the diagnosis, prevention, and management of heredo-familial diseases.

## Figures and Tables

**Figure 1 cancers-13-04807-f001:**
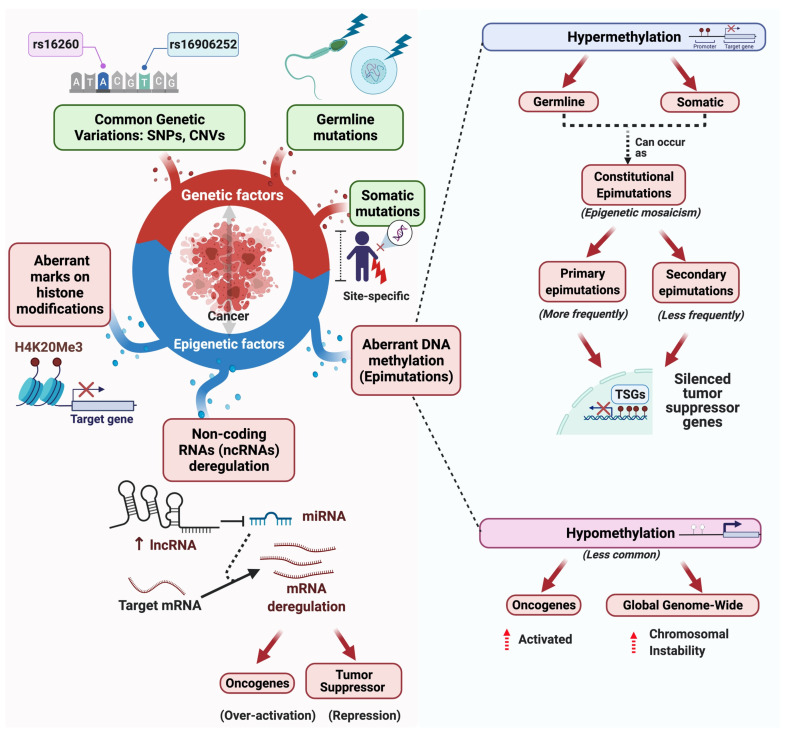
Contribution of genetic and epigenetic factors (epimutations) to cancer development. Cancer is caused by genetic factors such as genetic variations, somatic, and germline mutations and by epigenetic factors such as non-coding RNA dysregulation, post-translational alterations in histone marks, and aberrant DNA methylation (epimutations). DNA hypermethylation is the most frequent epimutation and can be generated either from the germline or somatic cells. In germline epimutations (before fertilization), all cells in the body are affected, while in somatic mutations (after fertilization) epigenetic mosaicism can be generated.

**Figure 2 cancers-13-04807-f002:**
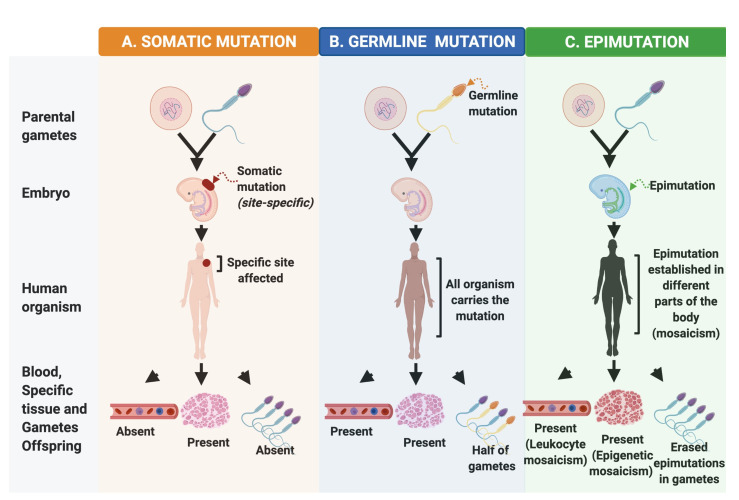
Early alterations during human carcinogenic development. Three important events for early carcinogenic development are described: (**A**) somatic mutations, (**B**) germline mutations, and (**C**) epimutations; focused on the different stages from the parental gametes, embryonic, in the human body, and within the latter detected as the presence or absence of the alteration in blood, specific tissue, and descendant gametes.

**Figure 3 cancers-13-04807-f003:**
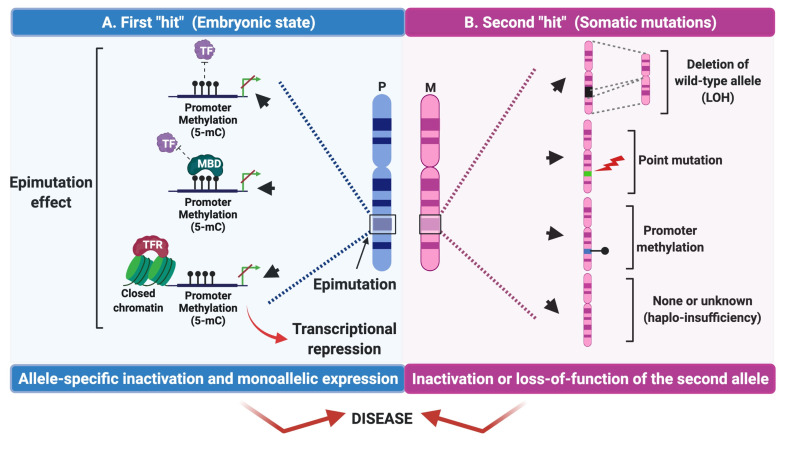
Knudson’s “two-hit hypothesis” on epimutations and cancer. (**A**) The first hit is promoted by epimutations in the promoter region in germ cells. This prevents the binding of TFs to DNA (top). Another mechanism dependent on MBD can be recruited into methylated DNA and compete directly with TFs through their greater affinity for the 5-mC site to the DNA sequence (middle part), or that act synergistically with TFRs to transcriptionally repress a gene. In all cases, this first hit induces the inactivation of the mono-allelic expression of a gene. (**B**) The second hit is driven by somatic cell mutations such as deletions (LOH), point mutations, other allelic epimutations, or by haploinsufficiency mechanisms that trigger the loss-of-function of the second allele (biallelic inactivation) and induce tumorigenesis. P: Paternal, M: Maternal, TF: Transcription Factor, TFR: Transcription Factor Repressor, MBD: Methyl-CpG-binding domain (MBD), 5-mC: 5-Methylcytosine, LOH: Loss of Heterozygosity.

**Figure 4 cancers-13-04807-f004:**
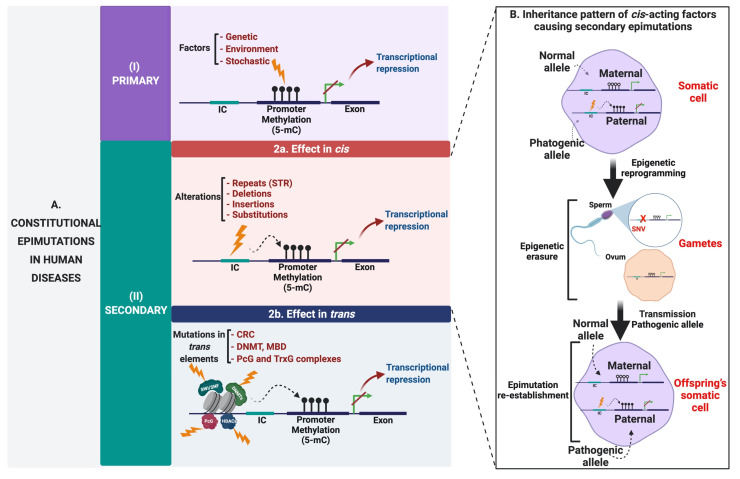
Constitutional epimutations in human diseases. (**A**) Constitutional epimutations that cause disease in humans include mechanisms that can occur without altering the DNA sequence (primary epimutations) or as a consequence of a DNA mutation in a *cis-* or trans-acting factor (secondary epimutations). (**I**) Primary epimutations result as a consequence of the action of different factors that directly modify the methylation pattern in the promoter region of a tumor suppressor gene. (**II**) Secondary epimutations that act in cis (**2a**) may be a consequence of alterations that affect the conserved parental region IC and generate hypermethylation of the promoter region. (**2b**) Secondary trans-acting factor epimutations occur as a result of mutations in trans elements such as CRC, DNMT’s, MBD, and PcG/TrxG complexes, which indirectly lead to an aberrant methylation state in the promoter region of the gene. In all cases, this leads to transcriptional repression of the tumor suppressor gene. (**B**) The transmission of the pathogenic allele occurs dominantly (in this example the paternal allele) starting from an alteration in *cis* (secondary epimutation in *cis*) in somatic cells. However, there is an erasure of the epimutation (epigenetic erasure) in the gametes, but the alteration in *cis* is held (in this example an SNV), which can be transmitted to the offspring while the epimutation is restored in somatic cells, being able to generate epigenetic mosaicism. IC: Imprinting Center, DNMT: DNA methyltransferase, MBD: Methyl-CpG-binding domain (MBD), 5-mC: 5 Methylcytosine, CRC: Chromatin-Remodeling Complex, PcG: Polycomb group, TrxG: Trithorax group (TrxG).

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
