# Peer review of "Cis-Acting Factors Causing Secondary Epimutations: Impact on the Risk for Cancer and Other Diseases"

_cancers, 2021, doi:10.3390/cancers13194807_

Round 1

Reviewer 1 Report

The authors submitted a review about secondary epimutations (that correspond to hypermethylation caused by cis-inherited mutations) and provide evidence concerning their impact on cancer with a focus on how this can be used in clinical screenings in order to reduce the mortality of such heritable cancers.

The article starts with a very well detailed graphical abstract about how and when appear DNA methylation during the lifetime of an individual.

They first introduce the epigenetic mechanisms (Figure1). Then they report the actual knowledge about the origin and consequences of primary and secondary epimutations, and the text is graphically resumed in figure 2, with details about differences between germline, somatic and epimutations that arise during the development. Then they fill the gap between epigenetic changes and cancer and particularly how does heritability impact epigenetic changes. They makes the link with the environment (Figure 3) and emphasize on the incidence of germline background on secondary epimutations (Figure 4). They also propose an interesting figure (5) that details chronologically all the reported evidences about the role of secondary epimutations in cancer and other diseases, from 1997 to 2018.

To summarize, the paper is well constructed, the figures are detailed and help the reader to follow the text, they cite around 150 previous papers and they detailed chronologically corresponding reports. For all these reasons I will encourage the publication in its present form.

Author Response

Thank you very much for giving a broad and detailed overview of our work and your support about the construction and illustration of the review. We appreciate your kind comments about this manuscript.

Reviewer 2 Report

This review from Ruiz de la Cruz et al. nicely summarizes the impact of secondary epimutations on some inherited diseases. Overall, I believe that the review is comprehensive, is well-written and very well aided by the figures, and helps the readers have a complete overview of the field.

I have some comments and suggestions for the authors:

  1. The title made me think that the majority of the text would be focused on cancer. However, it is more or less 50% cancer and 50% other diseases. Regarding cancer, since the authors focused only on familial cancer, I would suggest a slight change in the title, to avoid highlighting cancer: “Cis-acting factors causing secondary epimutations: impact on the risk of inherited diseases”.

  1. Similarly, at line 12 of the abstract, I think that it should state: “… and present evidence of their impact on inherited diseases”.

  1. I understand that the focus of the review are epimutations, therefore it makes sense to describe inherited cancer. But epigenetic alterations have been described to be also essential in sporadic cancer. Please, could you mention something about it?

  1. Again, I see the focus on cis-epimutations, but I think that epigenetics involves many complex mechanisms that constantly interplay, and cannot be ignored to understand the whole process. Maybe you could add a last section to integrate this information in the context of other epigenetic alterations –as at the beginning of the introduction, when ncRNAs and post-translational modifications of histones are mentioned.
    1. By the way, in the 3rd paragraph of the introduction, I missed any mention to histone acetylation and methylation, which are probably the two most studied PTMs described in histones affecting gene expression.

  1. In the 2nd paragraph of the 2nd section, it is stated that “epimutations are epigenetic alterations that can occur at specifics stages in normal cells, for example, when a tumor suppressor gene is methylated during the early stages of embryonic development”. It made me think in epigenetic events, mainly changes in methylation, which are needed for tissue differentiation during embryonic development in a physiological context, not in disease. Please, would you add some sentences to explain how this process is differentially regulated in disease and normal states?

  1. Page 8, last line: “the methylation of CpG cytosines is surprisingly variable in hypermethylated epigenotypes obtained from a single individual”. Why is it surprising? Methylation per se is variable, it is not a yes/no event. Although the promoter region of a gene is methylated at somewhat extent, it can be still expressed. Often there is a strong negative correlation between the amount of methylation and the amount of expression. Traditional techniques such as MSP used to provide a qualitative result –methylated/unmethylated. ”Novel” techniques such as pyrosequencing offer a quantitative point of view of methylation, and allow us to calculate cut-off points to better discriminate between hypomethylated and hypermethylated forms of a gene. In my opinion, the variation mentioned by the authors is not surprising, even in the context of cancer, where all pathological cells are supposed to have a clonal origin. Thus, it is even less surprising in non-cancerous diseases. Please, explain this better if I misunderstood.

  1. In this regards, Figure 5 is a very nice way to summarize all this information. I appreciate that the authors added the “method” column. Please, could you add more information in the text about how the method to detect methylation can influence the knowledge we currently have?
    1. Also, in this figure I did not find the TP53

  1. Figure 4B: I think that the middle panel should state “Gametes”, instead of “Offspring’s gametes”. The epigenetic erasure takes place in gametes (also in offspring’s gametes, of course), but following the logical sequence of time, in the upper panel we should have a somatic cell from a given individual, his/her gametes in the middle panel, and offspring’s somatic cells in the bottom panel. Please, confirm this.

  1. Page 13, 3rd paragraph, line 7: what is a “compound heterozygotes”?

  1. I found some minor issues with abbreviations:
    1. Page 6, last paragraph: “ICs” was already explained above
    2. Figure 3 legend: definition of “TFs” is included at the end, so you do not need to explain it in the 2nd Also, definition of TFR should be place at the end, along with the other abbreviations.
    3. Page 10, last paragraph, 2nd line: CLL should be stated instead of CCL.

Author Response

The reviewer’s observations were really useful and very well detailed, which allowed us to concisely address all the suggestions. We thank you for your considerations. Here, we present the resolution to each comment.

  1. The title made me think that the majority of the text would be focused on cancer. However, it is more or less 50% cancer and 50% other diseases. Regarding cancer, since the authors focused only on familial cancer, I would suggest a slight change in the title, to avoid highlighting cancer: “Cis-acting factors causing secondary epimutations: impact on the risk of inherited diseases”.

Answer: We thank the reviewer for this suggestion. We appreciate this comment, however, we believe that the title accurately reflects the manuscript content, considering that we cover eight cancer types and four inherited syndromes.

  1. Similarly, at line 12 of the abstract, I think that it should state: “… and present evidence of their impact on inherited diseases”.

Answer: We appreciate this suggestion, but we consider that the exclusion of the cancer diseases covered in the manuscript would reduce the accuracy of the information.

  1. I understand that the focus of the review are epimutations, therefore it makes sense to describe inherited cancer. But epigenetic alterations have been described to be also essential in sporadic cancer. Please, could you mention something about it?

Answer: Thank you for this comment. We added meaningful evidence of important epimutations in some sporadic cancers, regarding different molecules that drive epigenetic changes. This change can be found in section 2. “Origin and consequences of epimutations”. Paragraph 5. Page 6.

  1. Again, I see the focus on cis-epimutations, but I think that epigenetics involves many complex mechanisms that constantly interplay, and cannot be ignored to understand the whole process. Maybe you could add a last section to integrate this information in the context of other epigenetic alterations –as at the beginning of the introduction, when ncRNAs and post-translational modifications of histones are mentioned.

Answer: This is an important point raised by the reviewer. We recognize that diverse epigenetic mechanisms are of utmost importance in the biological behavior of cancer and may act in concert. To address this point we included additional information about ncRNAs in the introduction section 1, paragraph 2, page 2, and we also refer the reader to selected more comprehensive reviews on these topics.

  1. By the way, in the 3rd paragraph of the introduction, I missed any mention to histone acetylation and methylation, which are probably the two most studied PTMs described in histones affecting gene expression.

Answer: We agree with the reviewer. Somehow, we missed to mention both PTMs, indeed. We added post-translational modifications of histones "acetylation and methylation". This change can be found in section 1. “Introduction”. Paragraph 3. Page 2.

  1. In the 2nd paragraph of the 2nd section, it is stated that “epimutations are epigenetic alterations that can occur at specifics stages in normal cells, for example, when a tumor suppressor gene is methylated during the early stages of embryonic development”. It made me think in epigenetic events, mainly changes in methylation, which are needed for tissue differentiation during embryonic development in a physiological context, not in disease. Please, would you add some sentences to explain how this process is differentially regulated in disease and normal states?

Answer: We complemented in more detail the importance of methylation during early embryonic development under normal conditions and when epimutations can be generated at each stage and cause diseases. This change can be found in section 2. “Origin and consequences of epimutations”. Paragraph 4. Page 6.

  1. Page 8, last line: “the methylation of CpG cytosines is surprisingly variable in hypermethylated epigenotypes obtained from a single individual”. Why is it surprising? Methylation per se is variable, it is not a yes / no event. Although the promoter region of a gene is methylated at somewhat extent, it can be still expressed. Often there is a strong negative correlation between the amount of methylation and the amount of expression. Traditional techniques such as MSP used to provide a qualitative result –methylated / unmethylated. "Novel" techniques such as pyrosequencing offer a quantitative point of view of methylation, and allow us to calculate cut-off points to better discriminate between hypomethylated and hypermethylated forms of a gene. In my opinion, the variation mentioned by the authors is not surprising, even in the context of cancer, where all pathological cells are supposed to have a clonal origin. Thus, it is even less surprising in non-cancerous diseases. Please explain this better if I misunderstood.

Answer: Thank you. We appreciate your feedback. In fact, we commited an error in the writing of this paragraph. We have corrected this statement. What we wanted to express is that the study showed that an increase in CGG repeats in the patient with fragile X syndrome (FXS) is associated with the hypermethylation of the promoter region of the FMR1 gene. You can find this change in section 6. “Cis-acting factors causing secondary epimutations: historical evidence”. Paragraph 1. Page 10.

  1. In this regards, Figure 5 is a very nice way to summarize all this information. I appreciate that the authors added the “method” column. Please, could you add more information in the text about how the method to detect methylation can influence the knowledge we currently have?

Answer: We added more information regarding the methods to detect methylation and some advantages when using the current application. This change can be found in section 1. “Introduction”. Paragraph 4. Page 3.

  1. Also, in this figure I did not find the TP53.

Answer: We appreciate this comment. However, the TP53 gene is not presented in figure 5 since cis-acting secondary epimutations have not been described as an oncogenic mechanism in this gene.

  1. Figure 4B: I think that the middle panel should state “Gametes”, instead of “Offspring’s gametes”. The epigenetic erasure takes place in gametes (also in offspring's gametes, of course), but following the logical sequence of time, in the upper panel we should have a somatic cell from a given individual, his / her gametes in the middle panel, and offspring's somatic cells in the bottom panel. Please confirm this.

Answer: We thank you for pointing this out. We took your observation into account and changed “Offspring’s gametes” to “Gametes” in Figure 4B and its figure legend. This change can be found in Figure 4. Page 9.

  1. Page 13, 3rd paragraph, line 7: what is a “compound heterozygotes”?

Answer: We considered your observation and changed “compound heterozygotes” to only “heterozygotes”, since we wanted to imply that they are heterozygous for the genetic mutation. You can find this change in section 6. “Cis-acting factors causing secondary epimutations: historical evidence”. Paragraph 11. Page 14.

  1. I found some minor issues with abbreviations:

  1. Page 6, last paragraph: “ICs” was already explained above.

Answer: We corrected "ICs" in the paragraph and eliminated the definition as it was already described. This change can be found in section 4. “Primary epimutations and environmental factors”.          

  1. Figure 3 legend: definition of “TFs” is included at the end, so you do not need to explain it in the 2nd Also, definition of TFR should be place at the end, along with the other abbreviations.

Answer: Thank you. The legend of Figure 3 was modified by removing the definition of “TFs” in the text, since it is already described at the end. We also put the definition of “TFR” at the end. This change can be found in Figure 3. Page 8.

  1. Page 10, last paragraph, 2nd line: CLL should be stated instead of CCL.

Answer: Thank you very much for notice this mistake. We changed the abbreviation error "CCL" to "CLL". You can find this change in section 6. “Cis-acting factors causing secondary epimutations: historical evidence”. Paragraph 6. Page 11.

Reviewer 3 Report

A review report for the manuscript entitled “Cis-acting factors causing secondary epimutations: impact on the risk for cancer and other diseases” by Cruz and colleagues (cancers-1310128)

This is a review paper describing epimutation that cause cancers, mainly focusing previous studies for secondary aberrant DNA methylations caused by primary genetic mutations.  The manuscript is generally well written and concisely explains causal mechanisms and consequences of previously reported cancer-related epimutations.  I have only a few concerns in this manuscript as listed below.

(1) Erasure of DNA methylation in the germ line.

The authors describe that active mechanisms erase genomic DNA methylation in the germ line (Graphical Abstract, etc.).  However, it is known that this process most likely occurs mainly through passive mechanisms coupled with DNA replication in primordial germ cells during the migration to genital ridges, even though there are contributions of active mechanisms (for review, for example, ”Germ cell reprogramming”, Kurimoto & Saitou, Curr Top Dev Biol. 2019;135:91-125).  Thus, the existence of passive DNA demethylation could be included in the manuscript.

(2) DNA methylation.

In this manuscript, only the methylation of CpG dinucleotide is referred to as DNA methylation.  However, another DNA methylation, N6-adenosine methylation, has been also found in mammals, and could be mentioned in the manuscript.

(3) “Polycomb/trithorax complexes”

In this manuscript, Polycomb and Trithorax complexes are repeatedly referred to as “Polycomb/trithorax complexes”.  However, this is confusing, because these complexes are composed of different proteins with different enzymatic activities that cause opposite gene regulations.  In addition, if you saw “Polycomb/trithorax complexes” and “SWI/SNF complexes” in the same manuscript, you might be confused as if Polycomb and Trithorax are components of one protein complex.

(4) “Consequently, alterations in this regulatory mechanism lead to an aberrant increase (hypermethylation) or decrease (hypomethylation) in basal expression levels that have been associated with diseases such as cardiovascular problems [33,34], mental disorders [35,36], and different types of cancer [37-39].” (page 3, the end of the second paragraph)

“basal expression levels” is confusing, because “expression level” is usually used for gene expression level but not DNA methylation level.

(5) “Recently, it’s has been reported …” (page 3, last paragraph)

May be “Recently, it has been reported …”?

(6) “The genomic-specific phenotypes of printing …” (page 4, first paragraph)

May be “The genomic-specific phenotypes of imprinting …”?

(7) “HBA2 upstream deletion (CpG island),” (page 10, the end of the second paragraph)

Deletion occurs downstream of the HBA2 gene in Figure 5C.

(8) “This EPCAM deletion includes the stop codon that results in a continuous transcription, leading to an EPCAM-MSH2 fusion transcript and transcriptional repression of MSH2 [131,132].”

Deletion of the stop codon may cause an error of translational termination, but not that of transcriptional termination.  In this case, deletion of polyadenylation signal may be a cause of the continuous transcription (as is described in ref. 131 in the manuscript). 

Author Response

We thank you for your observations. Your feedback let us strengthen our knowledge on this subject. Here are the answers to each of your concerns.

  1. Erasure of DNA methylation in the germ line.

The authors describe that active mechanisms erase genomic DNA methylation in the germ line (Graphical Abstract, etc.). However, it is known that this process most likely occurs mainly through passive mechanisms coupled with DNA replication in primordial germ cells during the migration to genital ridges, even though there are contributions of active mechanisms (for review, for example, ”Germ cell reprogramming” , Kurimoto & Saitou, Curr Top Dev Biol. 2019; 135: 91-125). Thus, the existence of passive DNA demethylation could be included in the manuscript.

Answer: Thank you for this observation. To address this comment, we added  more information about this phenomenon during the early stages of early embryonic development. This change can be found in section 2. “Origin and consequences of epimutations”. Paragraph 4. Page 6.

  1. DNA methylation.

In this manuscript, only the methylation of CpG dinucleotide is referred to as DNA methylation. However, another DNA methylation, N6-adenosine methylation, has also been found in mammals, and could be mentioned in the manuscript.

Answer: Thanks for raising this point. Information on N6-adenosine DNA methylation in mammals was added to the text. This change can be found in section 1. “Introduction”. Paragraph 5. Page 3.

  1. "Polycomb / trithorax complexes"

In this manuscript, Polycomb and Trithorax complexes are repeatedly referred to as “Polycomb / trithorax complexes”. However, this is confusing, because these complexes are composed of different proteins with different enzymatic activities that cause opposite gene regulations. In addition, if you saw “Polycomb/trithorax complexes” and “SWI/SNF complexes” in the same manuscript, you might be confused as if Polycomb and Trithorax are components of one protein complex.

Answer: We agree with the reviewer on this point. Both compounds were separated in all the text to avoid confusion. We replaced "Polycomb/trithorax complexes" for "Polycomb (PcG) and Trithorax (TrxG) complexes" and we specified each one’s role. This change can be found in section 1. “Introduction”. Paragraph 3. Page 2. Also, in Figure 4A. (2b) and in its legend at the end. Page 10.

  1. “Consequently, alterations in this regulatory mechanism lead to an aberrant increase (hypermethylation) or decrease (hypomethylation) in basal expression levels that have been associated with diseases such as cardiovascular problems [33,34], mental disorders [35,36] , and different types of cancer [37-39]. " (page 3, the end of the second paragraph) "basal expression levels" is confusing, because "expression level" is usually used for gene expression level but not DNA methylation level.

Answer: Thank you for this comment. To avoid the confusion, we replaced “basal expression levels” with “basal levels” which is used at DNA methylation levels. This change can be found in section 1. “Introduction”. Paragraph 5. Page 3.

  1. “Recently, it’s has been reported…” (page 3, last paragraph) May be “Recently, it has been reported…”?

Answer: The statement “Recently, it’s has been reported” was replaced by “Recently, it has been reported”. This change can be found in section 2. “Origin and consequences of epimutations”. Paragraph 2. Page 5.

  1. “The genomic-specific phenotypes of printing…” (page 4, first paragraph May be “The genomic-specific phenotypes of imprinting…”?

Answer: Thank you for this comment. The statement "The genomic-specific phenotypes of printing ..." was replaced by "The genomic-specific phenotypes of imprinting ...". This change can be found in section 2. “Origin and consequences of epimutations”. Paragraph 3. Page 5.

  1. HBA2 upstream deletion (CpG island),” (page 10, the end of the second paragraph). Deletion occurs downstream of the HBA2 gene in Figure 5C.

Answer: Indeed, the deletion occurs downstream of the HBA2 gene. However, we meant that the hypermethylation of the CpG island occurs upstream from the site of the deletion. To avoid confusion, we changed wording in this paragraph. Now it is more clear. You can find this change in section 6. “Cis-acting factors causing secondary epimutations: historical evidence”. Paragraph 3. Page 11.

  1. "This EPCAM deletion includes the stop codon that results in a continuous transcription, leading to an EPCAM-MSH2 fusion transcript and transcriptional repression of MSH2 [131,132]."

Deletion of the stop codon may cause an error of translational termination, but not that of transcriptional termination. In this case, deletion of polyadenylation signal may be a cause of the continuous transcription (as is described in ref. 131 in the manuscript).

Answer: Thank you very much for the accuracy of this suggestion. We replaced "This EPCAM deletion includes the stop codon that results in a continuous transcription…” by “This EPCAM deletion includes the polyadenylation signal that results in a continuous transcription…”. You can find this change in section 6. “Cis-acting factors causing secondary epimutations: historical evidence”. Paragraph 8. Page 13.